# Nutrient Status among Latvian Children with Phenylketonuria

**DOI:** 10.3390/children10060936

**Published:** 2023-05-26

**Authors:** Olga Lubina, Linda Gailite, Julija Borodulina, Madara Auzenbaha

**Affiliations:** 1Children’s Clinical University Hospital, 1004 Riga, Latvia; madara.auzenbaha@rsu.lv; 2Scientific Laboratory of Molecular Genetics, Riga Stradinš University, 1007 Riga, Latvia; linda.gailite@rsu.lv (L.G.); juliaborodulina@inbox.lv (J.B.); 3European Reference Network for Hereditary Metabolic Disorder, 1004 Riga, Latvia

**Keywords:** phenylketonuria, nutrients, food diary, nutritional assessment, deficiency

## Abstract

(1) Introduction: Phenylketonuria (PKU) is an autosomal recessive inborn error of phenylalanine metabolism. The main treatment for PKU is to manage nutrition, thereby restricting phenylalanine intake. Part of patient management is analyzing eating habits to substitute missing nutrients and limit the overdose of nutrients. This is mainly done by analyzing food diaries. This is the first review of Latvian PKU patients eating habits performed by analyzing 72-h food diaries (FD). (2) Materials and Methods: This study included individuals between the ages of 18 and 31 years, PKU patients and 31 age- and sex-matched control groups. All respondents kept 72-h food diaries (FD) and underwent testing for zinc, selenium and ferritin levels in the blood. Food diary data were analyzed by Nutritics software to calculate the theoretical intake of nutrients, and these values were compared with the Ministry of Health of the Republic of Latvia’s recommended values. (3) Results: A lack of motivation and diet therapy compliance in PKU patients was observed during this research. A total of 32% of PKU patients refused to fill out their FD or filled it out incorrectly. The analysis of nutrient intake was observed, and there was a statistically significant difference between PKU patients in the 1–3 age group and the control group in fat intake. Fat intake in PKU patients was below MRHL recommendations. The intake of iron was found to be surplus in all PKU patients in the age group of 1–3, 91% of PKU patients in the age group of 4–6 years, 63% in the age group of 7–12 and 71% in the 13–18 year age group. Although there were no instances in the PKU patients nor the control group who had ferritin levels above the normal range. Selenium intake was surplus in 80% of PKU patients in the 1–3 age group, 91% in PKU patients in the 4–6 age group, 88% in the 7–12 age group and 86% in the 13–18 age group. None of the patients had Se levels in the blood above the normal range. Zn intake was surplus in 100% of PKU patients in the 1–3 age group, 82% in PKU patients in the 4–6 age group, 88% in the 7–12 age group and 57% in the 13–18 age group, and no PKU patients had high Zn levels. None of the control group participants had levels below the normal range of Zn and Se while 11% of PKU patients in the 13–18 age group had inadequate levels of Se, although Se intake based on their FD was optimal. (4) Conclusions: Regular PKU patient nutritional status evaluation is important to define and prevent possible nutrient deficiency, and further investigation should be continued to find out the mechanism of nutrient absorption in PKU patients. To prevent macronutrient deficiency such as fat and micronutrient deficiency in PKU patients, one could use supplements or try an improved nutrient content of Phe-free formula in the future.

## 1. Introduction

Phenylketonuria (PKU) is an autosomal recessive genetic disorder of amino acid metabolism found in Europe [1]. It is caused by a deficiency of phenylalanine hydroxylase in the liver and is usually diagnosed during newborn screenings. The main treatment for PKU is nutrition management, specifically the restriction of natural phenylalanine (Phe) in the diet in combination with special Phe-free protein substitutes based on L-amino acids, which effectively reduce the concentration of Phe in the blood [2]. However, strict diets with a very low intake of Phe can lead to nutrient deficiency and cause health issues such as osteopenia and cardiovascular diseases [3,4]. Micronutrients such as selenium, zinc, iron, calcium, poly-unsaturated fatty acids and vitamin D are at risk of deficiency due to the elimination diet [5]. The main components of a low-protein diet for PKU are fruit and vegetables that are naturally low in protein and Phe, special low-protein foods (SLPF) and Phe-free protein substitutes. Although SLPF are low in protein, they contain high amounts of carbohydrates (CHO) and fat [6]. Despite being vital to combating this disorder, a very strict low-protein diet can negatively affect a patient’s quality of life and consequently lower their adherence to the diet [7,8]. Nutritional assessment is included in the minimum requirements for the management and follow-up of patients with PKU defined in the European Society for Phenylketonuria and Allied Disorders Treated as Phenylketonuria (E.S.PKU) guidelines. Indeed, a clinical nutritional assessment should be conducted at every outpatient clinic visit, including a 24-h recall food record or 72-h food diary (FD) [1].

Although PKU is a well-known disorder and an extensive amount of research on the nutrient intake and eating habits of PKU patients exists [1,4,9], this is the first study to focus on Latvian PKU patients combining self-reported nutrition intake with objective evaluation using biochemical analysis of micronutrients important for brain development such as selenium, zinc and ferritin (referring to iron intake).

The aim of this study was to evaluate the nutritional status of children with PKU using 72-h FD, biochemical measurements of zinc, selenium and ferritin and to analyze how it compares with the recommendations outlined by the Ministry of Health of the Republic of Latvia (MHRL). The last step is to compare the data with results from the control group, due to a high risk of a deficiency in micronutrients such as zinc, selenium and iron in PKU patients due to a strict low-protein diet.

## 2. Materials and Methods

### 2.1. Place and Time of the Study

The study was performed at the Clinic of Medical Genetics and Prenatal Diagnostics, Latvian Rare Disease Coordination Centre, Children`s Clinical University Hospital, Riga, Latvia during 2019–2020.

### 2.2. Characteristics of the Study Population

In 2020, Latvia had 116 registered PKU patients, 29 of which were unable to have follow-ups due to a variety of reasons (death, emigration and non-compliance to PKU treatment). A total of 83 patients received dietary treatments, and five are on sapropterin dihydrochloride treatment (a pharmaceutical formulation of tetrahydrobiopterin (BH4)). Forty-six patients were under 18 years of age, and 37 were adults over the age of 18. In the study, only individuals under 18 years of age were included as it is very important to define potential nutrient deficiency in childhood and to minimize the possibility of the risk of developing a diet-related disorder. During the study, none of the PKU patients or control group individuals got diagnosed any other medical conditions and didn’t receive any other medication (including antibiotics).

### 2.3. Inclusion and Exclusion Criteria

Study inclusion criteria:Patients under 18 years of age;Patients diagnosed with the inherited metabolic disorder—phenylketonuria.

Exclusion criteria:Patients above 18 years of age;Patients or their parents who refused to complete food diaries.

### 2.4. Study Procedures and Methods

During their outpatient appointments over one year, all pediatric patients under 18 years were asked to complete a detailed 72-h FD (a record of all food and beverages consumed within this time period) and bring it to their next appointment or send it to the dietician via e-mail. Information on food brand names and cooking and preparation methods was included, and portion sizes were to be reported as accurately as possible using common unit sizes or preferably by weighing the food.

Patients were also asked to have blood tests to analyze their levels of zinc, selenium and ferritin. Venous blood was obtained from fasting subjects. Zinc was analyzed using photometric methods: for selenium, nuclear mass spectrometry was used and for ferritin, analysis immune fermentative chemiluminescence methods were used.

Patients were divided into four age groups (1–3 years, 4–6 years, 7–12 years, and 13–18 years) to compare our data on food intake analysis and blood test results with the recommended nutrient daily intake values for healthy populations set by the Ministry of Health of the Republic of Latvia’s.

Anthropometric parameters such as height and weight were measured in patients during follow-up visits every three months and analyzed using World Health Organization growth percentile charts. Body mass index (BMI) was calculated using anthropometric data [10].

### 2.5. Data Analysis

The 72-h FD data were analyzed using a nutrient content database created from the particulars of local and foreign food nutrients and Nutritics Nutritional Software (v5.81) [11]. The Nutritics database included the nutrient analysis of protein substitutes, special low protein foods, special local foods and information supplied by the manufactures. If some information was missed, local retail chain websites with food descriptions of nutrient content was used.

The parameters analyzed included the following: intake per day of total energy (measured in kilocalories (kcal)), total protein, natural Phe, amino acid supplements, fat, CHO and common vitamins and minerals. If a FD was not completed according to the instructions or information was missing (missing data about the amount of a product, unknown food brand), it was excluded from our analyses.

The acquired data was then compared with 31 age- and sex-matched individuals in a healthy control group (without known chronic illness or dietary restriction). Control group individuals were invited to participate in this study using social media. These individuals kept a 72-h FD and performed blood tests as described above which were appraised in relation to the MHRL recommended values for the daily intake of nutrients for healthy populations. [12].

### 2.6. Ethics

Written informed consent was obtained from all participating individuals or their legal guardians. The study was performed according to the Declaration of Helsinki, and the protocol was approved by the Central Medical Ethics Committee of Latvia (Nr.1/19-02-11).

## 3. Results

Of the 83 PKU patients, all 46 pediatric patients (49.4%) were asked to complete a 72-h FD. Fifteen patients refused to complete a FD, and two FDs were invalid due to missing information on the precise amounts of food or contained beverages incompatible with the dietician’s instructions. Therefore, 31 of the 46 FDs (67,39%) were suitable for the food intake analysis.

Of these 31 patients, five were under the age of three, 11 were in the 4–6 age group, eight were in the 7–12 age group, and seven were in the 13–18 age group. Fifteen girls and 16 boys kept diaries and did blood tests (See Table 1).

To evaluate the body composition of PKU patients and the control group, their body mass index (BMI) was used. It was found that obesity was more common in female PKU patients, and being overweight was more common among male PKU patients (general overview given in Table 1). None of the parameter cases or the PKU group had statistically significant differences (*p* > 0.05).

### 3.1. Analysis of Nutrient Intake

Total nutrient intake was analyzed within the age groups and then compared with the recommended nutrient intake per day for the Latvian population and healthy control group. These results are shown in Table 2.

The total protein intake in the PKU group was assessed, including the amounts of protein from amino acid supplementation. Statistically significant differences were observed between PKU patients and the control group according to the total fat intake in the age group of 1–3 years (*p* < 0.004).

Results of the share of nutrient in the total energy intake is presented in Table 3.

Furthermore, by evaluating the total protein intake and natural Phe intake in the different age groups against the E.S.PKU recommended values, it was found that Latvian PKU patients consumed less protein than the recommended amount in the 7–12 years age group. In adolescents, there was no difference between the recommended and consumed total protein intakes.

### 3.2. Micronutrient Status

For our PKU patients, by analyzing the data from the 72-h FD, we found that their intakes of selenium (Se), zinc (Zn) and iron (Fe) were even higher than the values recommended by the MHRL and higher than the micronutrient intake of the control group (Table 4). The table shows results from the 72-h FD and biochemical tests.

#### 3.2.1. Biochemical Testing for the Zinc, Ferritin and Selenium Levels

Test results are shown in Table 4.

#### 3.2.2. Selenium

When Phe-free protein substitutes were excluded from the nutrient intake analysis, all patients except for one were found to have insufficient total Se. The exception has a mild form of PKU and can tolerate more natural protein than the other patients. Two patients out of the five were undergoing BH4 therapy and were found to have insufficient intake of iron and zinc. These three patients all had a surplus intake of selenium.

#### 3.2.3. Zinc

The daily intake of Zn was also fully dependent on Phe-free protein substitute intake in PKU patients and in the control group. The main source of Zn was meat, poultry, dairy products and grains such as oats.

When Phe-free protein substitutes were excluded from the analysis, Zn deficiency was identified in 20 patients, while one patient had an adequate Zn intake, and another had a higher intake (9 mg) than the RDI (7 mg).

Zn levels in PKU patients in the 1–3 age group were within the normal range (71%), 50% in the 4–6 age group, 80% in the 7–12 age group and 50% in the 13–18 age group. Levels below the normal range of Zn were mainly in the 4–6 and 13–18 age groups (50% equally). None of the PKU patients had results above the normal range compared with the control group where a surplus of zinc levels was found. A total of 60% of the female PKU patients had low Zn levels in the 4–6 and 13–18 age groups.

#### 3.2.4. Iron

The daily intake of Fe was also fully dependent on Phe-free protein substitute intake. The main sources of iron in a healthy diet are meat, poultry, eggs and fish products.

#### 3.2.5. Ferritin

None of the PKU patients and the control group respondents showed a surplus in ferritin levels. A total of 20% of male PKU patients in the 4–6 age group and 17% of male PKU patients in the 7–12 age group were in deficit. A total of 25% of female PKU patients in the 1–3 age group, 20% in the 4–6 age group and 17% in the 13–18 age group were in a deficit.

#### 3.2.6. Summary

Iron intake was surplus in all PKU patients in the 1–3 age group, 91% of PKU patients in the 4–6 age group, 63% in the 7–12 age group and 71% in the 13–18 age group. None of the PKU patients nor the control patents had ferritin levels above the normal range.

Selenium intake was surplus in 80% of PKU patients in the 1–3 age group, 91% of PKU patients in the 4–6 age group, 88% in the 7–12 age group and 86% in the 13–18 age group, and none of the patients had blood Se levels above the normal range.

Zn intake was surplus in 100% of PKU patients in the 1–3 age group, 82% in the 4–6 age group, 88% in the 7–12 age group and 57% in the 13–18 age group, and none of the PKU patients had high levels of Zn.

None of the control group participants had levels below the normal range of Zn and Se, and 11% of PKU patients in the 13–18 age group had inadequate levels of Se, although Se intake based on their FDs was optimal.

Zn intake based on FD data was below the MHRL recommended values in 43% of the PKU patients in the 13–18 age group, and 50% of them also had a Zn deficit based on their blood test results.

## 4. Discussion

Dietary phenylalanine restriction is currently a main treatment method, although there are other options such as sapropterin dihydrochloride and enzyme substitution therapy with pegvaliase. Nucleic acid therapy (therapeutic mRNA or gene therapy) is likely to provide longer term solutions with few side effects. However, the cost of such therapies represents a new challenge that healthcare systems will need to address [13].

This is the first review of pediatric PKU patients’ nutrient status and dietary intake in Latvia. Our focus was to evaluate macro and micronutrients such as zinc, selenium and iron intake and status due to the high possibility of deficiency. Many PKU centers around the world, including ours, report a lack of adherence to therapeutic diets during adolescence and adulthood. As a result, it is difficult to engage with patients to complete an FD, and it is particularly difficult to motivate them to record precise amounts and the types of food they consume during the day; thus, the main limitation of this study was the small amount of usable data. Our evaluation of the nutritional status of PKU patients was restricted by a limited number of completed FDs. Our analysis was further hampered by the absence of a united database of the contents of Latvian foods and a lack of information on the nutritional composition of SLPF, as reported previously [6]. Almost one-third of pediatric PKU patients did not complete a FD or completed it in incorrectly. In 2018, Cazzorla et al. confirmed that metabolic control and compliance with dietary treatments are poor in adult PKU patients [8]. Poor adherence in adolescent and adult PKU patients was also reported by Jurecki et al. who observed a suboptimal adherence to the recommended Phe concentrations [14]. At present, there is a significant lack of information on the nutritional status of Latvian adolescent and adult (over the age of 18) PKU patients [15]. The reason for this is their non-compliance with completing a 72-h FD. Therefore, to address this gap in our knowledge, we need to regularly assess their nutritional intake using 24-h recall food records during outpatient clinic visits [14,16].

Lack of adherence brings us to another limitation of this study, which was self-reported information in the FD. Unfortunately, we cannot always trust the data that are provided in the FD because patients may not provide accurate information. In this study, we found discrepancies in some of the information provided in the FD. For example, patients, whose BMI was above the 95th percentile provided a dietitian-filled FD, and there was an observed lack of calories and nutrients; therefore, real energy and nutrient intake was under-reported. As found earlier, the metabolic control of Latvian PKU patients becomes worse with age. In some patients, their Phe levels were above 360 mmol/l; therefore, their intake of natural protein could not be as low as detailed in their FD. [15] Additionally, inconsistencies were seen with BMI and FD data on CHO and total energy intakes. Specifically, three obese patients reported CHO and total energy intakes that were lower than the values recommended by the MHRL. Also, other research groups reported that children have been found to under-report caloric intake by as much as 40% [17].

We observed that most of the patients reported a CHO intake that was above the MHRL’s recommended level. This is in line with the findings of Sailer et al. [18] and Couce et al. showing that PKU patients had a higher energy intake from CHO than the control group [19]. In contrast, the fat intake of the vast majority of our PKU patients was below the MHRL’s recommended level. A low intake of fat is a characteristic finding of many PKU patients studies. For example, a study by Rose et al. in 2005 concluded that the youngest children (<5 years of age) in their group of PKU patients appeared to be especially vulnerable to inadequate fat intake due to a lack of diversity in their diets [20]. These results are in accordance with our data showing that the biggest disparity between recommended and actual fat intake was observed in the 1–3 age group. Furthermore, data presented by Schulpis et al. and van Calcar et al. revealed that PKU diets tend to contain more CHO and less fat (20–25% energy) compared to a typical omnivore diet that provides 30–35% energy from fat [21,22]. It has been proposed that all PKU patients diagnosed with a nutrient imbalance should undergo dietary corrections [23]. Our future research will investigate the type and quality of fat intake and its correlation to lipid and fatty acid status in serum.

The Phe-free formula is fortified with many micronutrients, including Se, Zn and iron. This is probably the reason why deficiency of these micronutrients was not observed in the FD (biochemical investigations confirmed it) of our PKU patients. However, when we excluded the mixture of amino acids from the nutrient intake analysis, the patients were then found to have a severe deficiency of these nutrients. Se deficiency has been reported in many PKU patient studies. For example, Okano et al. concluded that Se intake in Japan was low and suggested that this was due to a lack of Se in Phe-free protein substitutes available in the country [9]. Thus, the bio-availability of Se derived from the Phe-free formula requires further investigation. The most common natural source of Se for Latvian PKU patients is egg yolk and bread [24]. This is on the yellow list of products and so should be avoided or restricted to a very small amount per day [25].

In our study, we see that the Zn intake is within the normal range, although the blood Zn levels are below the normal range by 50% in the age groups of 4–6 and 13–18 years. Zn deficiency may be due to a low Zn bioavailability. Diets with high fiber, phytates and other mineral content may affect the Zn bioavailability as reported in Baines et al. [26]. A similar Zn deficiency in the blood was observed in the study presented by Barretto et al. [27]. We will continue to assess patient’s food diaries and nutrient status in the blood to improve the provided dietary treatment of our patients.

## 5. Conclusions

The continuous evaluation of the nutritional status of PKU patients is important for diagnosing and treating possible nutrient deficiencies. Regular intake of Phe-free protein substitutes in accordance with the recommendations of a dietician is important not only for controlling Phe levels in the blood but also for ensuring the appropriate intake of energy, fat, protein, Se and Zn. Although Phe-free protein substitutes contain the primary vitamins and minerals, there is still uncertainty regarding their levels of absorption and utilization in PKU patients. We found that our PKU patients had a significantly lower intake of fat compared to the MHRL’s recommended level. This is why we must encourage PKU patients to increase healthy fat intake, for example adding olive oil, canola oil, walnut oil, avocado and other nut and seed oils.

## Figures and Tables

**Table 1 children-10-00936-t001:** Description of PKU and control groups by sex, age and BMI.

	PKU	Control
Male/female ratio	16/15	16/15
Age, years ± SD	7.8 ± 4.9	8.3 ± 5.1
Weight, kg ± SD	29.8 ± 20	30.6 ± 18
Height, m ± SD	1.16 ± 0.29	1.27 ± 0.31
BMI, kg/m^2^ ± SD	17.3 ± 3.4	17.2 ± 2.1
Obesity, %	13%	7%
Overweight, %	16%	10%
Normal, %	61%	79%
Underweight, %	10%	0%
Severe underweight, %	0%	3%

PKU—phenylketonuria, SD—standard deviation, BMI—body mass index.

**Table 2 children-10-00936-t002:** Nutrient intake based on FD in PKU patients and the control group compared with suggested values from the Ministry of Health of the Republic of Latvia.

	Total Energy Intake (kcal/day) Median ± SD	Total Protein Intake (g/day) Median ± SD	Total CHO Intake (g/day) Median ± SD	Total Fat Intake (g/Day) Median ± SD
*PKU*				
1–3 years (*n* = 5)	1236 ± 403	42 ± 12	210 ± 40	27 ± 16
*p*-value *	0.738	0.738	0.738	0.004
4–6 years (*n* = 11)	1142 ± 371	41 ± 14	191 ± 90	29 ± 14
*p*-value *	0.705	0.964	0.923	0.063
7–12 years (*n* = 8)	1963 ± 541	60 ± 29	260 ± 83	55 ± 43
*p*-value *	0.175	0.657	0.238	0.657
13–18 years (*n* = 7)	1682 ± 394	64 ± 17	213 ± 93	48 ± 17
*p*-value *	0.462	0.269	0.086	0.217
*Control group*				
1–3 years (*n* = 5)	973 ± 206	44 ± 8	122 ± 49	40 ± 4
*p*-value *	0.738	0.738	0.738	1.000
4–6 years (*n* = 11)	1461 ± 276	55 ± 11	184 ± 48	49 ± 7
*p*-value *	0.786	0.242	0.365	0.997
7–12 years (*n* = 8)	1870 ± 241	75 ± 61	232 ± 79	66 ± 7
*p*-value *	0.985	0.762	0.972	0.762
13–18 years (*n* = 7)	2051 ± 565	59 ± 16	229 ± 68	76 ± 39
*p*-value *	1.000	1.000	0.995	0.979
*MHRL*				
1–3 years	960–1290	24–48	108–194	32–57
4–6 years	1200–1433	30–54	135–215	40–56
7–12 years	1400–1830	35–92	158–275	47–71
13–18 years	1920–2283	48–114	216–342	64–89

PKU—phenylketonuria, MHRL—Ministry of Health of the Republic of Latvia, CHO—carbohydrate, * *p*-value—calculated using Fisher’s Exact Test one-tailed *p*-value comparing each group with MHRL suggested values.

**Table 3 children-10-00936-t003:** Nutrient share in energy intake in the analyzed case control group.

	Fat Share in Energy Intake	vs. MHRL	Protein Equivalent Share in Energy Intake	vs. MHRL	CHO Share in Energy Intake	vs. MHRL
PKU						
1–3 years	20%	−50%	13%	−10%	68%	13%
4–6 years	23%	−35%	14%	−4%	67%	12%
7–12 years	25%	−28%	12%	−39%	53%	−12%
13–18 years	26%	−26%	15%	−24%	51%	−16%
Control group						
1–3 years	37%	−7%	18%	20%	50%	−16%
4–6 years	30%	−14%	15%	0%	50%	−16%
7–12 years	32%	−10%	16%	−20%	50%	−17%
13–18 years	33%	−4%	12%	−42%	45%	−26%

**Table 4 children-10-00936-t004:** Characterization of zinc, ferritin and selenium levels in analyzed individuals.

	Zinc		Selenium		Iron	
	Blood Level µg/dL	Nutrient Intake * (mg/Day)	Blood Level µg/dL	Nutrient Intake * (µg/Day)	Blood Level µg/dL	Nutrient Intake * (mg/Day)
PKU						
1–3 years (*n* = 5)	71 ± 8	13 ± 5	81 ± 13	68 ± 25	27 ± 13	17 ± 5
*p*-value *	0.318	1.000	1.000	1.000	0.986	0.500
4–6 years (*n* = 11)	76 ± 19	18 ± 7	75 ± 13	59 ± 17	34 ± 48	19 ± 7
*p*-value *	0.278	0.214	1.000	1.000	0.882	0.313
7–12 years (*n* = 8)	81 ± 14	14 ± 6	73 ± 8	100 ± 42	33 ± 41	19 ± 6
*p*-value *	0.838	0.429	1.000	1.000	0.949	0.296
13–18 years (*n* = 7)	71 ± 11	14 ± 6	67 ± 13	86 ± 35	39 ± 23	25 ± 10
*p*-value *	0.120	1.000	0.563	1.000	0.995	0.462
Control group						
1–3 years (*n* = 5)	104 ± 13	7 ± 2	68 ± 7	57 ± 45	20 ± 6	6 ± 1
*p*-value *	1.000	1.000	1.000	0.500	0.203	1.000
4–6 years (*n* = 11)	107 ± 16	8 ± 1	60 ± 5	54 ± 10	22 ± 13	6 ± 2
*p*-value *	0.943	0.982	1.000	1.000	0.489	1.000
7–12 years (*n* = 8)	109 ± 18	10 ± 2	72 ± 11	99 ± 42	22 ± 11	11 ± 1
*p*-value *	0.559	1.000	1.000	1.000	0.360	0.949
13–18 years (*n* = 7)	99 ± 15	7 ± 4	58 ± 8	90 ± 29	21 ± 24	9 ± 5
*p*-value *	0.989	1.000	1.000	0.538	0.071	1.000
Reference						
1–3 years	70–120	5–7	50–150	20–24	20–200	7–9
4–6 years	70–120	6–8	50–150	24–30	20–200	7–9
7–12 years	70–120	7–9	50–150	32–39	20–200	9–11
13–18 years	70–120	8–10	50–150	41–50	20–200	10–12

* Calculated from 72-h food diaries using Nutritics software.

## Data Availability

Not applicable.

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
