# Peer review of "Nutrient Status among Latvian Children with Phenylketonuria"

_children, 2023, doi:10.3390/children10060936_

Round 1

Reviewer 1 Report

Your papers is well structured and it does build around the purpose that was meant for.

It is obvious that the statistic methods are crispand that you spent lots of time trying to get the best out of your study.

I would sugest adding some novelties into the text. For example the are hopes about introdcing a new drug specific for the treatment of PKU -  a small-molecule inhibitor of the phenylalanine transporter SLC6A19 that has the potential to be a first-in-class oral therapy used to treat any person with PKU.

Another point that could be added to your paper is the interference of day-by-day medication with PKU.

English is good ...just a liitle correction is needed.

Author Response

Dear Reviewer 1!

Thank You very much for Your review and all the comments that You have made, I really appreciate it!

Thank You for suggesting adding some novelties into text about new treatment methods, I did so: Dietary phenylalanine restriction still is main treatment method nowadays, although there are other options like sapropterin dihydrochloride and enzyme substitution therapy with pegvaliase. Nucleic acid therapy (therapeutic mRNA or gene therapy) is likely to provide longer term solutions with few side efects. However, the cost of such therapies represents a new challenge that healthcare systems will need to address. [13]

Another point that You mentioned was adding information about interference of medications that use PKU patients. I am really sorry that I haven`t mentioned it in paper before, thank You a lot for pointing on this error. I added information in part of Methodology: On the time of study none of PKU patients or control group individuals got diagnosed any other medical conditions and didn’t receive any other medication (including antibiotics).

I am really grateful for Your suggestions and recommendations regarding my article, I hope that I have answered to all Your questions, if not please, let me know and I will try to do my best!

Reviewer 2 Report

Dear Authors,

Your study is important, but it contains many limitations that should be improved.

The introduction provides a comprehensive overview of phenylketonuria (PKU), a genetic disorder of amino acid metabolism caused by phenylalanine hydroxylase deficiency. The mainstay of PKU treatment is nutrition management, which involves a strict low-protein diet and Phe-free protein substitutes to reduce the concentration of Phe in the blood. However, such a diet can lead to nutrient deficiencies, including micronutrients such as selenium, zinc, iron, calcium, polyunsaturated fatty acids, and vitamin D. The introduction highlights the importance of nutritional assessment in the management and follow-up of patients with PKU and the need to evaluate the nutritional status of children with PKU in Latvia, including self-reported nutrition intake and biochemical analysis of important micronutrients for brain development such as selenium, zinc, and ferritin. The study aims to compare the nutritional status of Latvian PKU patients with the recommendations outlined by the Ministry of Health of the Republic of Latvia and results from the control group to determine the risk of micronutrient deficiencies in PKU patients. The introduction provides a clear and concise summary of the background, rationale, and objectives of the study, with a focus on the significance of the research question.

The Materials and Methods section of the study seems well-structured and informative. However, I have noticed some errors and areas for improvement, which I have listed below:

Errors:

In line 82, it says "all paediatric patients were asked to complete a detailed 72-hour FD." However, in line 78, it states that only individuals under 18 years of age were included in the study. Therefore, it would be more accurate to say, "all paediatric patients under 18 years of age were asked to complete a detailed 72-hour FD."

In line 105, it says "which were appraised in relation to the MHRL recommended values." However, it is not clear what "MHRL" stands for. It should be clarified.

Areas for improvement:

It would be helpful to provide more information on how the nutrient content database was created, as well as the sources of local and foreign food nutrients used.

It would be useful to include more information on the blood tests conducted, such as the specific methods used to analyze zinc, selenium, and ferritin levels.

In line 99, it would be helpful to explain how BMI and growth percentile charts were used to analyze anthropometric parameters.

In line 102, it would be useful to provide more information on how the healthy control group was selected and matched with the patient group.

To improve the perception of reception, divide the chapter of material and methods into further sections:

2.1 Place and time of the study

2.2 Characteristics of the study population

2.3 Inclusion and exclusion criteria

2.4 Study procedures and methods -

2.5 Data analysis

2.6 Ethics

Overall, the Materials and Methods section provides a good overview of the study design and procedures. However, providing more detail on certain aspects and addressing the errors mentioned above would improve the clarity and accuracy of the section.

The findings indicate that iron intake was surplus in all age groups of PKU patients, while selenium intake was surplus in most age groups. On the other hand, zinc intake was found to be below the recommended values in some PKU patients, particularly in the 13-18 age group. None of the PKU patients had high levels of zinc or selenium in their blood, and none of the participants in the control group had levels below the normal range of these nutrients.

However, there are some issues with the methodology used in this study that should be addressed. The study only focuses on nutrient intake and blood levels, and does not take into account other factors that can affect nutrient absorption and utilization, such as medications and medical conditions.

Furthermore, the study presents some inconsistencies in its results. For example, while the study indicates that none of the PKU patients had high levels of zinc in their blood, it also reports that zinc intake was surplus in most age groups. This raises questions about the accuracy of the measurements and the interpretation of the results.

You have not demonstrated all the limitations of your study, which are primarily due to methodological limitations. Please complete them. Add strengths as well.

Overall, the conclusions drawn from the study emphasize the importance of continuously monitoring the nutritional status of PKU patients to identify and address possible nutrient deficiencies. The regular intake of Phe-free protein substitutes, as recommended by a dietician, is also highlighted as crucial not only for controlling Phe levels in the blood but also for ensuring adequate intake of energy, fat, protein, Se, and Zn.

However, there are a few errors in the statement. First, the sentence in line 273 should read "Regular intake of Phe-free protein substitutes," instead of "Phe-free protein substitutes." Additionally, the statement in line 276 about uncertainty regarding the absorption and utilization of vitamins and minerals in PKU patients requires further clarification as it is a complex issue.

Moreover, the finding that PKU patients had a lower intake of fat compared to the recommended level suggests that these patients may need to increase their healthy fat intake. The suggestions given in line 280 are appropriate, but it is important to remember that the specific recommendations for healthy fat intake should be made by a registered dietician or a healthcare professional based on an individual's needs.

Technical notes. Place periods after square brackets, not before. Adjust writing according to ACS Style. Customize tables according to MDPI template.

Overall, the study provides valuable insights into the nutrient status of Latvian children with PKU. However, the limitations and inconsistencies of the study should be considered when interpreting the results. Further research is needed to investigate the nutrient status of this population and to identify effective interventions to address any deficiencies or surpluses.

Please respond to my comments point by point.

Greetings

The article contains minor language errors, which I have covered in the comments. 

Author Response

Dear Reviewer!

Thank You very much for Your review and all the comments that You have made, I really appreciate it!

Point 1 In line 82, it says "all paediatric patients were asked to complete a detailed 72-hour FD." However, in line 78, it states that only individuals under 18 years of age were included in the study. Therefore, it would be more accurate to say, "all paediatric patients under 18 years of age were asked to complete a detailed 72-hour FD."

Response 1 Thank You for pointing on inaccuracy describing patient`s age, I corrected this as You recommended.

Point 2 In the line 105, it says "which were appraised in relation to the MHRL recommended values." However, it is not clear what "MHRL" stands for. It should be clarified.

Response 2 – I am sorry that I haven`t described properly what are the recommendations of nutrient daily intake for healthy population set by Ministry of Health of Latvia, thank You for finding this inaccuracy. Corrected: These individuals kept a 72-hour FD and performed blood tests as described above which were appraised in relation to the MHRL recommended values for the daily intake of nutrients for healthy population.

Can You suggest if I should add these recommendations to my article as attachment?

Point 3 It would be helpful to provide more information on how the nutrient content database was created, as well as the sources of local and foreign food nutrients used.

Response 3 Thank You for suggestion, I added information about database: The Nutritics database included the nutrient analysis of protein substitutes, special low protein foods and special local foods, information supplied by the manufactures, if some information was missed – local retail chain websites with foods description of nutrient content was used.

Point 4 It would be useful to include more information on the blood tests conducted, such as the specific methods used to analyze zinc, selenium, and ferritin levels.

Response 4 Based on Your recommendation I added information about methods used to analyse micronutrients: Patients were also asked to have blood tests to analyse their levels of zinc, selenium and ferritin. Venous blood was obtained from fasting subjects.  Zinc was analysed using photometric method, for selenium nuclear mass spectrometry and for ferritin analysis immune fermentative chemiluminescence methods were used.

Point 5 In line 99, it would be helpful to explain how BMI and growth percentile charts were used to analyze anthropometric parameters.

Response 5 Corrected: Anthropometric parameters such as height and weight were measured in patients during follow-up visits every three months and analysed using World Health Organisation growth percentile charts, body mass index (BMI) was calculated using data of anthropometric parameters.[10]

Point 6 In line 102, it would be useful to provide more information on how the healthy control group was selected and matched with the patient group.

Response 6 Corrected: The acquired data was then compared with 31 age and sex-matched individuals in a healthy control group (without known chronic illness or dietary restriction). Control group individuals were invited to participate in this study using social media.  

Point 7 To improve the perception of reception, divide the chapter of material and methods into further sections:

2.1 Place and time of the study

2.2 Characteristics of the study population

2.3 Inclusion and exclusion criteria

2.4 Study procedures and methods -

2.5 Data analysis

2.6 Ethics

Point 7 Thank You very much for your suggestion to change chapter of material and methods and divide it to sections. I did as You recommended.

Point 8 The findings indicate that iron intake was surplus in all age groups of PKU patients, while selenium intake was surplus in most age groups. On the other hand, zinc intake was found to be below the recommended values in some PKU patients, particularly in the 13-18 age group. None of the PKU patients had high levels of zinc or selenium in their blood, and none of the participants in the control group had levels below the normal range of these nutrients.

However, there are some issues with the methodology used in this study that should be addressed. The study only focuses on nutrient intake and blood levels, and does not take into account other factors that can affect nutrient absorption and utilization, such as medications and medical conditions.

Response 8 – I agree that we mostly focused on nutrient intake un blood levels, it mainly was due to the fact that on the moment of the study none of PKU patients had any other medical condition and no specific medication (for example, antibiotics) where used. Although we may look into other nutrient effect on absorption, for example, high calcium or fibre intake, that can affect iron absorption.

Point 9 Furthermore, the study presents some inconsistencies in its results. For example, while the study indicates that none of the PKU patients had high levels of zinc in their blood, it also reports that zinc intake was surplus in most age groups. This raises questions about the accuracy of the measurements and the interpretation of the results.

Response 9 According this point there might be other factors, that can affect zinc absorption despite of hight intake. Added to article: In our study we see that Zn intake is in normal range although in the blood Zn levels are below normal range in 50% in the age group of 4-6 and 13-18 years. Zn deficiency may be due to low Zn bioavailability. Diet with high fibre, phytates and other minerals content may affect Zn bioavailability as it is reported in Baines at al. [26], similar Zn deficiency in the blood was observed in the study presented by Barretto et al.[27]

Point 10 You have not demonstrated all the limitations of your study, which are primarily due to methodological limitations. Please complete them. Add strengths as well.

Response 10 Thank You a lot for pointing on my study weaknesses, I added limitations to discussion part as You recommended.

Point 11 Overall, the conclusions drawn from the study emphasize the importance of continuously monitoring the nutritional status of PKU patients to identify and address possible nutrient deficiencies. The regular intake of Phe-free protein substitutes, as recommended by a dietician, is also highlighted as crucial not only for controlling Phe levels in the blood but also for ensuring adequate intake of energy, fat, protein, Se, and Zn.

However, there are a few errors in the statement. First, the sentence in line 273 should read "Regular intake of Phe-free protein substitutes," instead of "Phe-free protein substitutes." Additionally, the statement in line 276 about uncertainty regarding the absorption and utilization of vitamins and minerals in PKU patients requires further clarification as it is a complex issue.

Response 11 – correction made.

Point 12 Moreover, the finding that PKU patients had a lower intake of fat compared to the recommended level suggests that these patients may need to increase their healthy fat intake. The suggestions given in line 280 are appropriate, but it is important to remember that the specific recommendations for healthy fat intake should be made by a registered dietician or a healthcare professional based on an individual's needs.

Response 12 – Totally agree with You, thank You.

Point 13 - Technical notes. Place periods after square brackets, not before. Adjust writing according to ACS Style. Customize tables according to MDPI template.

Response 13 – Corrections made.

I am really grateful for Your suggestions and recommendations regarding my article, I hope that I have answered to all Your questions, if not, please, let me know and I will try to do my best!

Round 2

Reviewer 2 Report

Dear Authors, Thank you for taking into account my comments, I think the work has greatly improved in quality.